

# 1 Anomalous Amide Proton Chemical Shifts as Signatures of Hydrogen
# 2 Bonding to Aromatic Sidechains

Kumaran Baskaran*[1], Colin W. Wilburn*[1], Jonathan R. Wedell[1], Leonardus M. I. Koharudin[2], Eldon L.
Ulrich[1], Adam D. Schuyler[1], Hamid R. Eghbalnia[1], Angela M. Gronenborn[2], and Jeffrey C. Hoch[1]
[1]Department of Molecular Biology and Biophysics, UConn Health, 263 Farmington Ave., Farmington, CT 06030-3305 USA
[2]Department of Structural Biology University of Pittsburgh School of Medicine 3501 Fifth Ave., BST3/Rm. 1050 Pittsburgh, PA
15260 USA
*Correspondence to*: Jeffrey C. Hoch (hoch@uchc.edu)
Dedicated to Professor Robert Kaptein on the occasion of his 80th birthday.
**Abstract.** Hydrogen bonding between an amide group and the p-π cloud of an aromatic ring was first identified in a protein in the
1980s. Subsequent surveys of high-resolution X-ray crystal structures found multiple instances, but their preponderance was
determined to be infrequent. Hydrogen atoms participating in a hydrogen bond to the p-π cloud of an aromatic ring are expected
to experience an upfield chemical shift arising from a shielding ring current shift. We survey the Biological Magnetic Resonance
Data Bank for amide hydrogens exhibiting unusual shifts as well as corroborating nuclear Overhauser effects between the amide
protons and ring protons. We find evidence that Trp residues are more likely to be involved in p-π hydrogen bonds than other
aromatic amino acids, whereas His residues are more likely to be involved in hydrogen bonds with a ring nitrogen acting as the
hydrogen acceptor. The p-π hydrogen bonds may be more abundant than previously believed. The inclusion in NMR structure
refinement protocols of shift effects in amide protons from aromatic side chains, or explicit hydrogen bond restraints between
amides and aromatic rings, could improve the local accuracy of side-chain orientations in solution NMR protein structures, but
their impact on global accuracy is likely be limited.

## 21 1 Introduction

In 1986, Perutz et al.(Levitt and Perutz, 1988) identified a putative hydrogen bond between an amino group of Asparagine and an
aromatic ring of a drug bound to hemoglobin. Similar observations of the π electrons of aromatic rings acting as acceptors for
hydrogen bonding have been reported before and since.(Klemperer et al., 1954; Mcphail and Sim, 1965; Knee et al., 1987) Later
in 1986, Burley  and Petsko (Burley and Petsko, 1986) surveyed 33 high resolution protein structures and found further evidence
of aromatic hydrogen bonds.  Tüchsen and Woodward (Tüchsen and Woodward, 1987) subsequently observed an upfield shift in
the Gly-37 NH and Asn-44 HN resonances due to a nearby Tyr-35 aromatic group.  The measurements from this study allowed
Levitt and Perutz (Perutz, 1993) to estimate that these interactions contribute around 3 kcal mol$^{-1}$ in stabilizing enthalpy, about
half as strong as a conventional hydrogen bond. Further evidence of such H-bonding came from the 2001 study by Brinkley and
Gupta (Brinkley and B., 2001) showing FTIR spectroscopic evidence for hydrogen bonding between alcohols and aromatic rings.
The ability of aromatic rings to engage in weakly polar CH- π interactions is well documented, with NMR data from Plevin et
al.(Plevin et al., 2010) in the form of weak scalar (J) couplings between methyl groups and atoms in aromatic rings providing direct
evidence of these interactions.  The study also included a survey of 183 X-ray structures and found 183 putative Me/π interactions.
Brandl et al.(Brandl et al., 2001) surveyed 1154 protein structures from the Protein Data Bank (PDB (Consortium, 2019) for C–H
π  H bonds and found 14,087 involving aromatic rings and satisfying their geometric criteria.  This is made all the more impressive





when considering that Levitt and Perutz report the partial charges on the C–H group are one third those on the N–H group (the
subject of this paper), suggesting that the interaction studied by Brandl et al. is correspondingly weaker. Another survey of note
was performed by Weiss et al. in 2010.(Weiss et al., 2001) This complete hydrogen bond analysis of two high resolution protein
structures from PDB found 50 C–H π and two (N,O)–H π bonds.
In addition to their ubiquity, there is some indication of the importance of these interactions. In a 1993 review, Perutz (Perutz,
1993) indicated the potentially wide-ranging importance of these interactions, particularly Armstrong et al.'s demonstration of
their role in stabilizing α-helices(Armstrong et al., 1993). There is also evidence that similar interactions play an important role in
protein-ligand complexes.(Panigrahi and Desiraju, 2007; Polverini et al., 2008)
Following the example of Tüchsen and Woodward (Tüchsen and Woodward, 1987) we seek to use NMR to provide corroborative
evidence of aromatic hydrogen bonds. In this paper, we survey the Biological Magnetic Resonance Bank (BMRB) for unusual
amide proton chemical shifts and amide-aromatic nuclear Overhauser effects.

Theoretical models for the geometrical dependence of the ring current shift include parameterization of quantum-mechanical(Haigh
and Mallion, 1979; Memory, 1963) calculations, semi-classical approximation using the Biot-Savart Law(Jackson, 1999) for the
field arising from current loops (Waugh and Fessenden, 1957; Jr. and Bovey, 1958), and a dipole approximation. For distances
from the ring center that are greater than 3 Å above the plane of the ring, and 5 Å in the plane of the ring, the theories all agree
well with a dipole approximation.(Hoch, 1983) The $(1-3\cos^2(\theta))/r^3$ geometrical dependence of the field arising from a magnetic
dipole (where θ is the angle between the vector from a proton to the aromatic ring center and the vector normal to the plane of the
ring) provides vivid explanation for cone separating upfield-shifted from down-field-shifted regions defined by θ=54.7° (Figure

55   1).

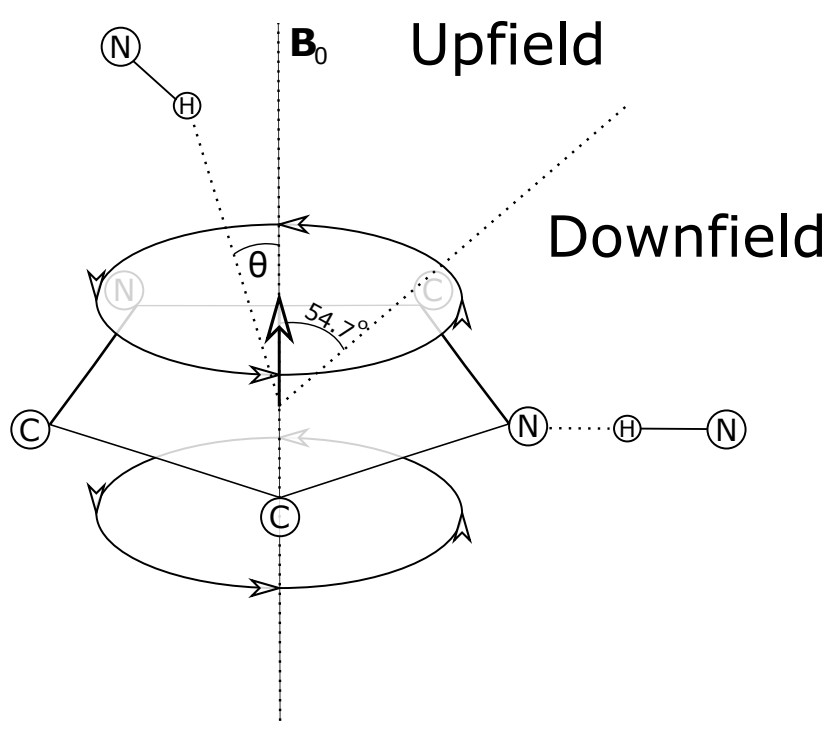






*Figure 1. Definition of the azimuthal angle (θ) and demarcation of regions of upfield and downfield ring current shifts.*
*For protons above the plane of a Tyr or Phe ring the upfield shift can reach 1.5 ppm for distances from the ring center around 3*
*Å; for protons in the plane of the ring the downfield shift approaches 2 ppm at 3 Å. For Trp the effects can be significantly larger.*
*Local mobility (e.g. fluctuations about the $\chi_2$ side-chain dihedral angle of the aromatic residue) can substantially diminish ring*
*current shifts.*[21]

**2 Approach**
To investigate the connection between amide proton chemical shifts and the potential for hydrogen bonding to an aromatic ring,
we searched BMRB for assigned amide protons in proteins corresponding to structures deposited in the PDB. BMRB provides the
list of BMRB and PDB entry id pairs via BMRB API (http://api.bmrb.io/v2/mappings/bmrb/pdb?match_type=exact) . As of Jan
2021 we found 7750 BMRB/PDB paired entries and retrieved the BMRB entries (in NMR-STAR format (Ulrich et al., 2019)) and
PDB entries (in mmCIF format (Bourne et al., 1997)) from their respective databases. We filtered out DNA/RNA entries, entries
with legends, oligomers and protein complexes. At the end we prepared a dataset consists of 363686 amide protons from 4670
entries. We combined the chemical shift information from BMRB and the geometric information form PDB for each amide proton
and its nearest aromatic ring using sequence number and residue name. For each assigned amide chemical shift, Z-score was
computed characterizing the deviation of the shift from its mean value from the BMRB database
$$Z = \frac{\delta_{res} - \bar{\delta}_{res}}{\sigma_{res}} \qquad\qquad (1)$$

where $\delta_{res}$ is the amide chemical shift of a given residue in ppm, $\bar{\delta}_{res}$ and $\sigma_{res}$ are the mean and the standard deviation of the amide
proton of a given residue type, based on statistics maintained by BMRB (https://bmrb.io/ref_info/stats.php?restype=aa&set=filt).
For each assigned amide, the distance from the amide position to the centre of the nearest aromatic ring is computed from the
coordinates in the PDB mmCIF file. The distance is defined as the average of the distance from the amide proton to the centre of
the aromatic ring, averaged over the members of the structural ensemble present in the PDB entry. For the nearest aromatic ring,
we calculated an azimuth angle (Figure 1), defined as the angle between a vector normal to the aromatic ring plane and the vector
between the amide proton and the centre of the ring. The ring normal vector is computed by calculating the cross product of two
vectors on the plane of the ring (say the vector from the centre of the ring to CG and CD1). The table of assigned chemical shifts,
Z-scores, distances to the nearest aromatic ring and azimuth angles is provided as a comma-separated text file (CSV file) in the
supplementary information. The workflow used in the analysis is depicted in Figure 2.

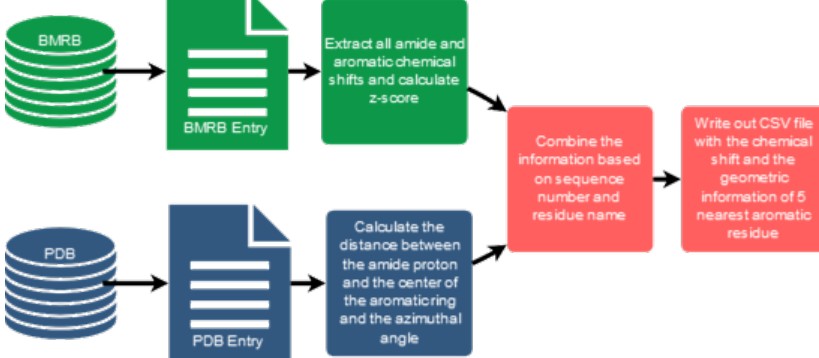




***Figure 2:*** *Manual federation of BMRB and PDB via a customized workflow.*
Corroboration of close proximity between an amide proton and an aromatic ring observed in PDB structures is found in assigned
distance restraints based on nuclear Overhauser effects (NOEs) present in the BMRB entries. NMR restraint files were downloaded
from the PDB and parsed using PyNMRSTAR (Smelter et al., 2017) for NOE restraints between amide protons and aromatic ring
protons of different residues. Because many files list NOEs under 'simple' distance restraints, these were included. Due to
inconsistencies prevalent in the restraint data, several criteria were implemented to ensure some conformity in the restraints
included in our analysis. This and other reasons for excluding entries from the restraints analysis are described in greater detail in
Supplementary Table 1. Also discarded were individual distance restraints which reported only a lower distance bound or an upper
distance bound greater than 6Å (as this is inconsistent with the nuclear Overhauser effect) and restraints that were ambiguously
between more than two different residues (in order to simplify the analysis). Of the entries that remained, 2564 listed at least one
restraint between an amide proton and an aromatic ring proton and 863 did not.
**3 Results and Discussion**
**3.1 Analysis of Chemical Shift Data**
Chemical shift Z-scores as a function of distance to the nearest aromatic ring are shown in Figure 3, separated by the type of
aromatic sidechain. For all four aromatic residue types, there is a clear correlation between proximity to the aromatic ring and the
amide chemical shift variance: significant deviations from the mean, corresponding to Z-scores greater than 2, are most likely
when the proton is proximal to an aromatic ring, and the magnitude of the shift deviations are larger for closer proximity. The
bottom row in Figure 3 examines the distribution of amide chemical shifts that are closer than 8 Å in greater detail.

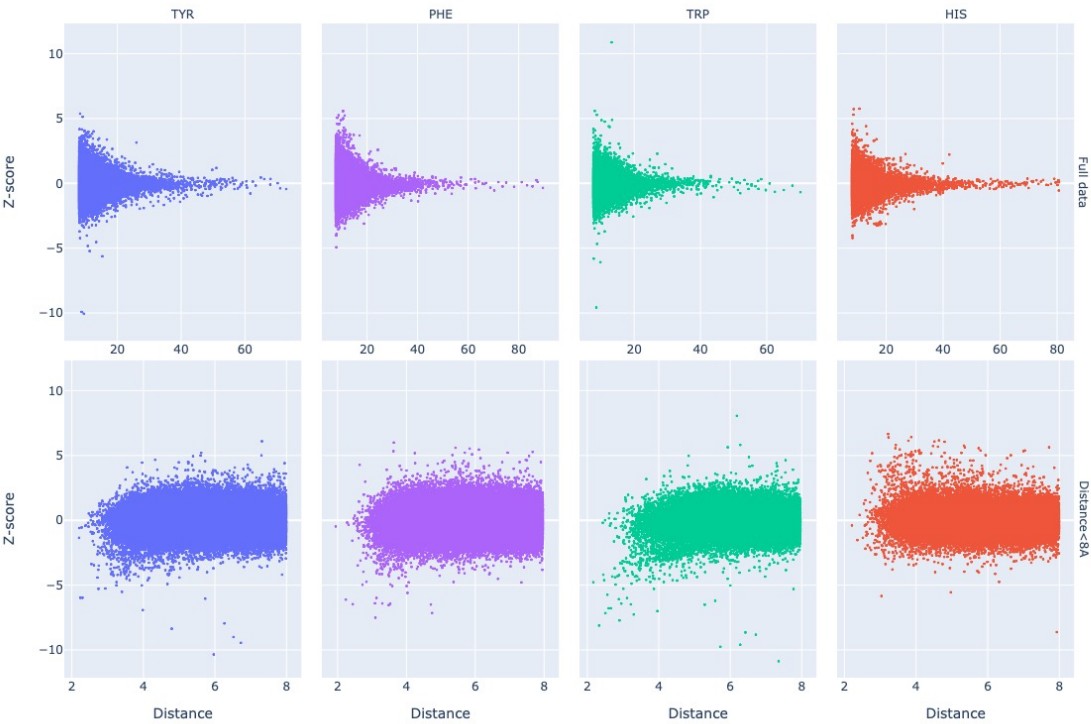






*Figure 3: The distribution of amide chemical shifts as function of distance from the center of the nearest ring.*

The figure illustrates differences in the pattern of chemical shift deviation for the four different types of aromatic sidechains. For
amide protons proximal to Phe, Tyr, or Trp sidechains, there is a noticeable preponderance of upfield shifts (negative Z-score). In
contrast, His amide protons exhibiting large deviations from the mean tend to be shifted downfield (positive Z-scores). The
difference in behavior of the outliers for the different aromatic residue types suggests the deviations are not simply the result of
residues buried in the protein interior. The upfield-shifted resonances for amides proximal to Phe, Tyr, and Trp are consistent with
hydrogen bonding between the amide and the p-$\pi$ electrons. The downfield-shifted resonances for amides proximal to His are
consistent with hydrogen bonding to the electronegative nitrogen atoms of the His ring. In-plane downfield ring current shifts are
the same sign as the expected downfield shifts arising from hydrogen bonding, with a predicted amide proton ring current shift of
0.5 ppm for an amide nitrogen distance of 3.4 Å. This is consistent with the observation of larger magnitude Z scores for downfield-
shifted amide protons proximal to His.

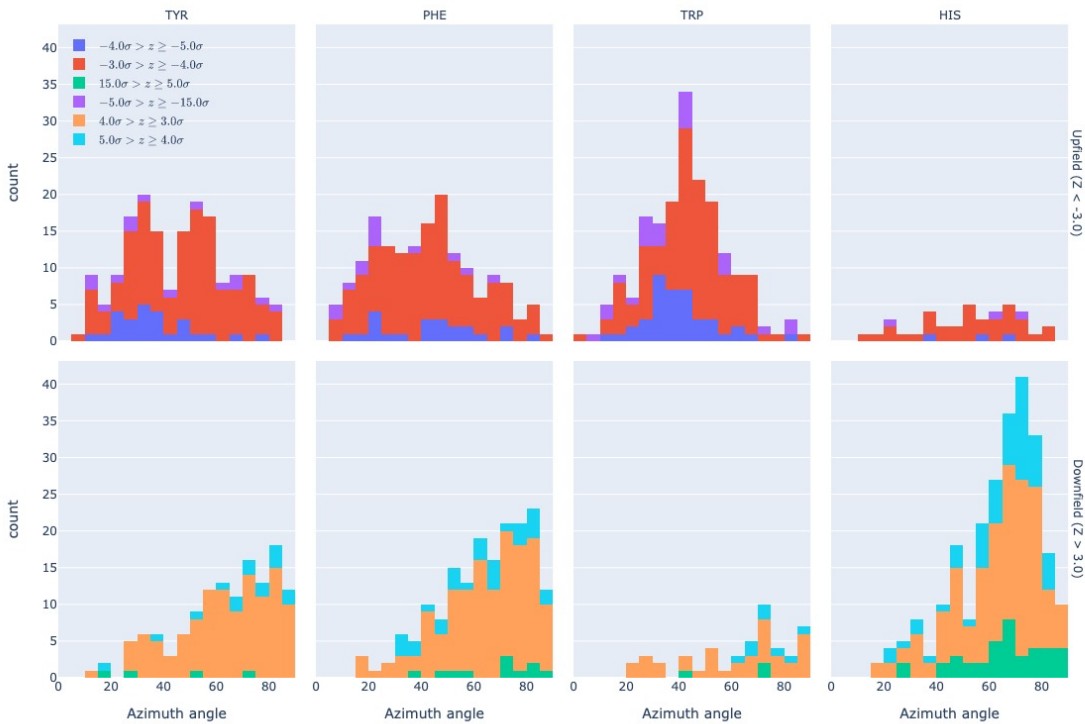


*Figure 4: Distribution of azimuth angles for outlier ($>3\sigma$) amide proton shifts. Upfield shifts are shown in the top row, downfield*
*shifts in the bottom row.*

Further evidence of the unusual behavior of amide protons with unusual shifts proximal to His and Trp residues is found in their
spatial distribution. Figure 4 shows the distribution of azimuth angle for upfield and downfield outliers that are within 8Å of an
aromatic ring. (Outliers are defined here as having absolute value of the Z-score greater than 3.) Shift outliers proximal to His tend
to reside near the ring plane, whereas shift outliers proximal to Trp tend to reside above the ring plane. Phe and Tyr don't exhibit
a pronounced preponderance of outliers above or near the ring plane.





**3.2 Analysis of Restraint Data**

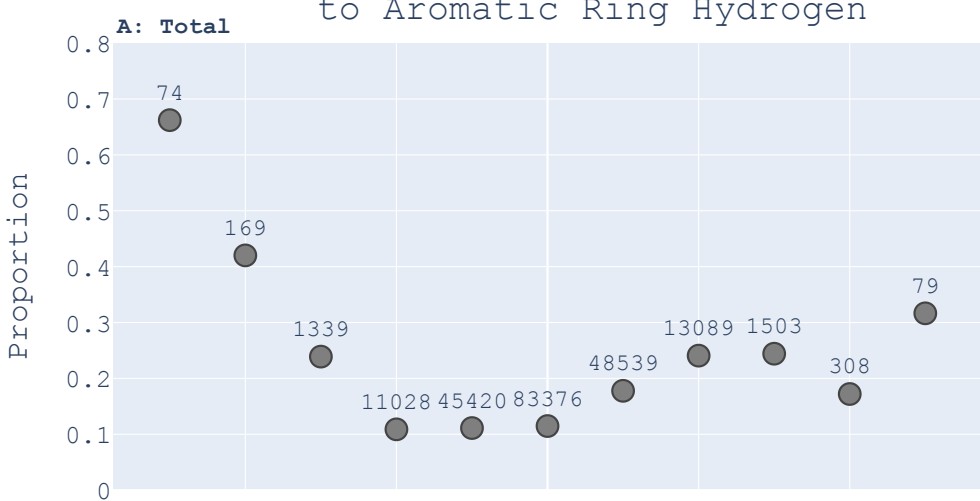

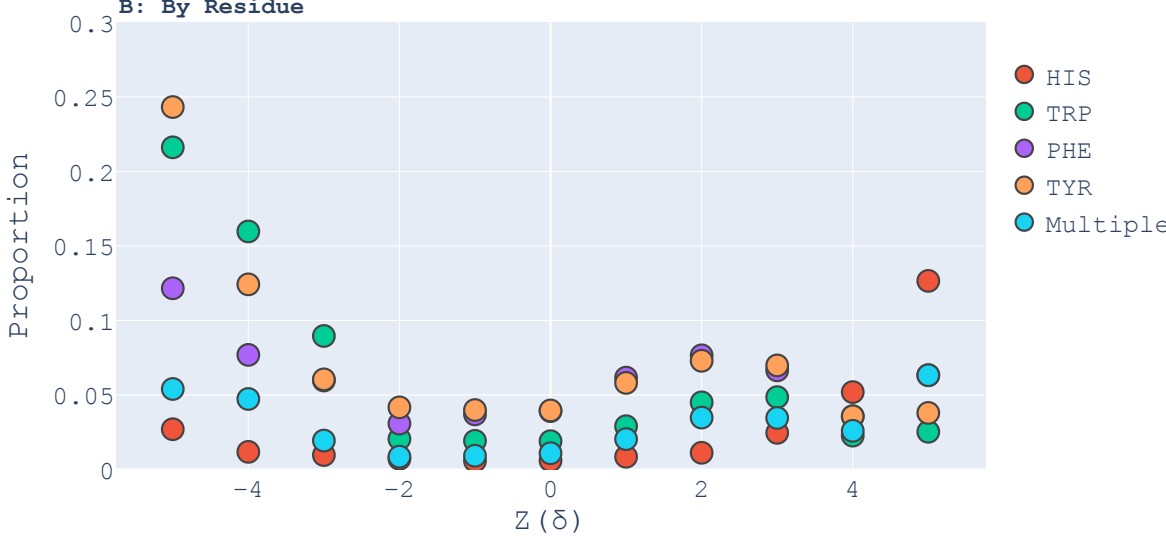


*Figure 5: Proportions of amide protons with at least one NOE restraint to an aromatic ring proton (y-axis), as a function of the*
*Z-score of the amide proton (x-axis). Proportions are calculated with respect to the total number of amide hydrogens with chemical*
*shifts reported in entries with at least one amide-aromatic restraint. The numbers over each point in panel **A** are the total number*
*of such amides (including those lacking any NOE restraints to a nearby aromatic) with that Z-score. In panel **B**, the restrained*
*amide protons are further demarcated by the type of aromatic sidechain to which they are restrained.*

We found 31,746 amide protons with at least one NOE restraint to a nearby aromatic ring. Figure 5A shows the proportion of
amide protons (from entries with usable restraint data and at least one amide-aromatic restraint) exhibiting these restraints. For



both upfield- and downfield-shifted amide protons, the greater the deviation from the mean the greater the likelihood that
corresponding NOE restraints are observed. The trend is noticeably more pronounced for the upfield-shifted amide protons, which
is consistent with the formation of hydrogen bonds between the amide and the p-$\pi$ electrons. The downfield-shifted amides exhibit
a weaker correlation, which may be indicative of other dominating effects (not necessarily due to nearby aromatic rings). Figure
5B further demarcates the data by the type of the nearby aromatic residue. We observe that the preponderance of amide-aromatic
restraints in upfield-shifted amide protons for interactions with Trp and Tyr (and to a lesser extent Phe). In contrast, amide protons
proximal to His residues predominate strong downfield shifts ($Z \geq 4$). This stands as further evidence for hydrogen bonding from
the amide to the p-$\pi$ electrons in Trp, Tyr, and Phe, and to the nitrogen atoms in the His ring.

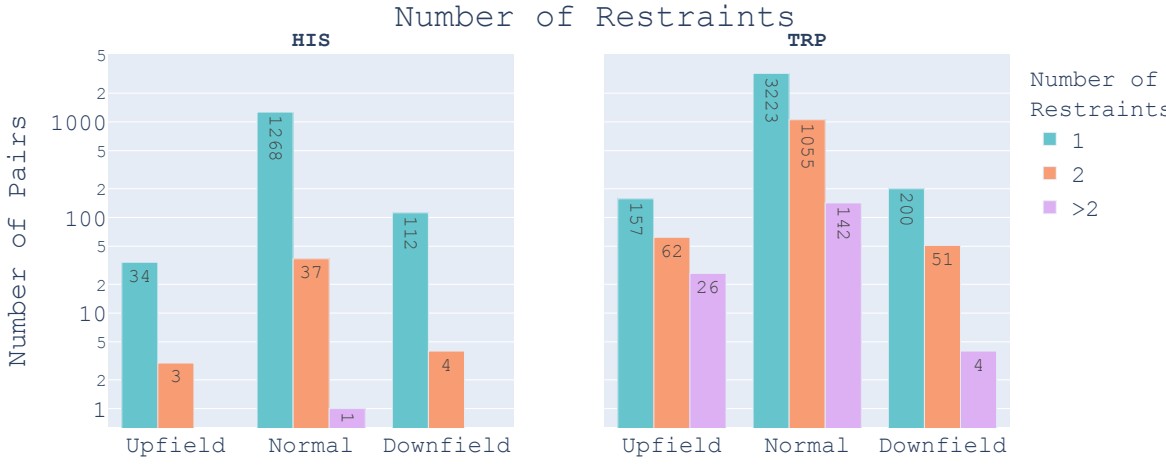

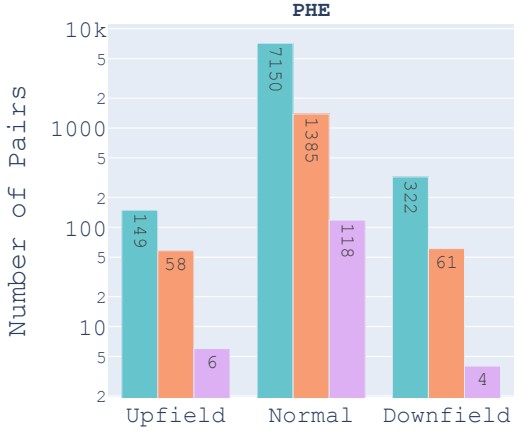

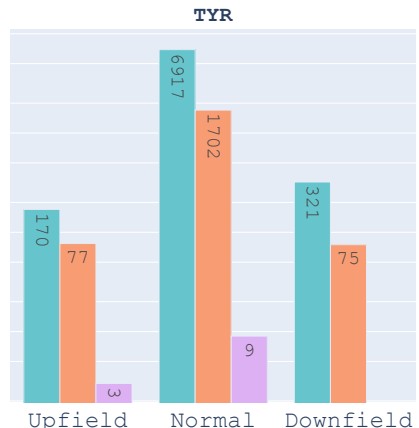




*Figure 6: Shown are the number of restrained amide-aromatic pairs (that is amide protons and aromatic rings with at least one*
*defined restraint between them) for the four aromatic residue types and three Z-score classifications: upfield (Z ≤ −2), downfield*
*(Z ≥ 2), and normal (−2 ≤ Z ≤ 2). The colors of the bars correspond to the number of restraints between the pairs.*

In Figure 6 the restrained amide-aromatic pairs are separated by the type of the aromatic residue and the number of restraints
between the amide proton and the aromatic ring protons. For every aromatic type, a greater proportion of the upfield-shifted pairs
have more than one restraint between them than the downfield-shifted pairs, which may indicate a hydrogen bond from the amide
to the p-π electrons. his observation is consistent with the others. Finally, the prevalence of restrained pairs with an outlier amide
is quite high. From the 2523 entries considered, 1166 such pairs were found, nearly one such pair in every two entries.
**3.3 Examples**

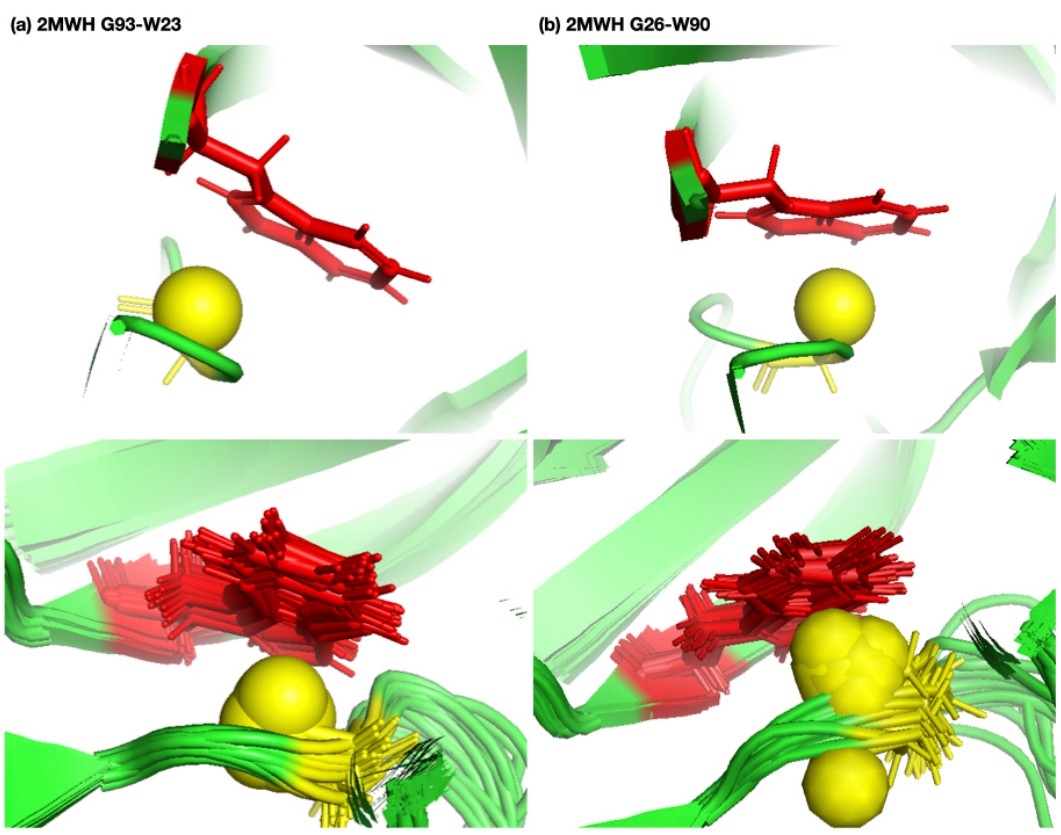


*Figure 7: Examples of amide protons with extreme upfield shifts  (a) PDB:2MWH The G93 amide proton is directly below the*
*W23 aromatic ring (Z =-7, $\delta_H$ = 2.937 ppm), (b) PDB:2MWH The G26 amide proton is directly below the W90 aromatic ring (Z*
*=-6.43, $\delta_H$ = 3.38 ppm) . The top row shows only the first model and the bottom row shows the ensemble. The amide proton is*
*represented as a yellow sphere and the aromatic side chain is shown in red*





Figure 7a & 7b shows the examples of p-$\pi$ hydrogen bond in the anti-HIV lectin Oscillatoria agardhii agglutinin (PDB ID:2MWH)
in which the amide chemical shifts of G93 (z-score =-7, $\delta_H$ = 2.937 ppm) and G26 (z-score =-6.43, $\delta_H$ = 3.38 ppm) are upfield
shifted due to the interaction of W23 and W90 respectively.

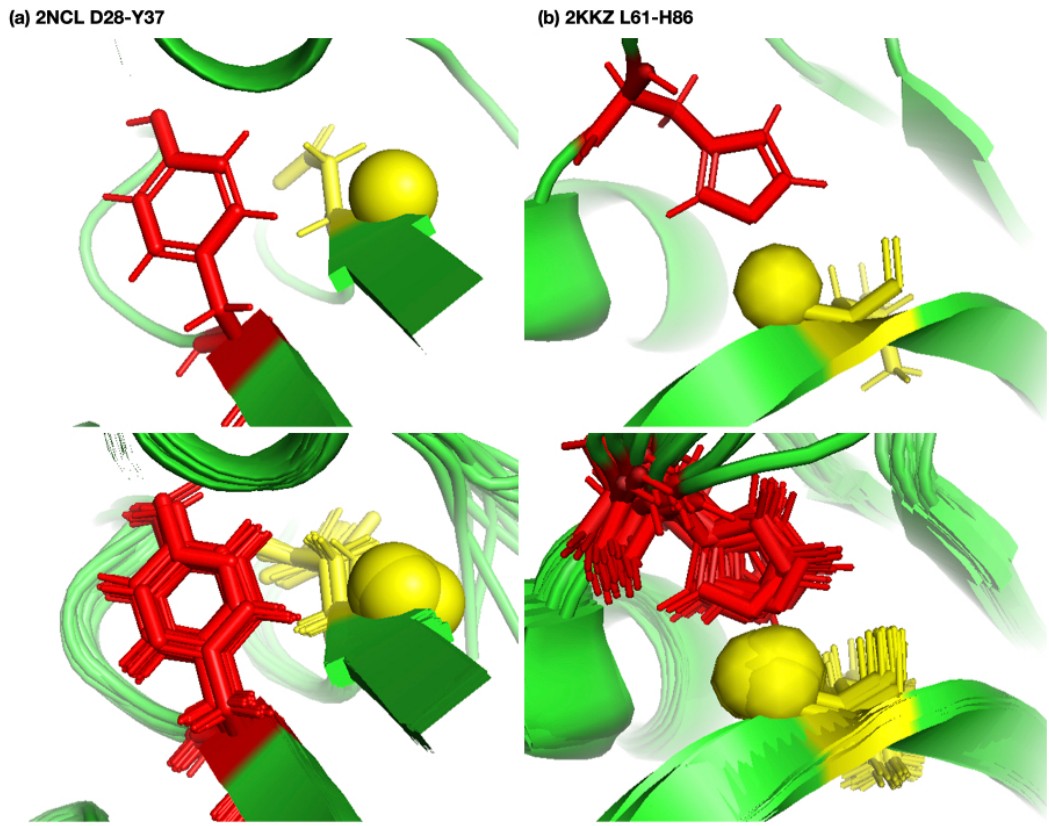


**Figure 8**: *Examples of amide protons with extreme downfield shifts. (a) PDB:2NCL The D28 amide proton is near the plane of*
*Y37 aromatic ring (Z =5.21, $\delta_H$ = 11.387 ppm),(b) PDB:2KKZ The L61 amide proton forms a hydrogen bond with the side chain*
*nitrogen of H86 (Z =6.66,$\delta_H$ = 12.66 ppm). The top row shows only the first model and the bottom row shows the ensemble. The*
*amide proton is represented as a yellow sphere and the aromatic side chain is shown in red.*


Figure 8a shows the amide proton of D28 is more of less on the plane of the Y37 aromatic ring in BOLA3 protein (PDB ID:2NCL)
resulting the amide chemical shift of D28(z-score =5.21, $\delta_H$ = 11.387 ppm) to shift downfield. Figure 8b shows an example of
possible hydrogen bond between the NE2 of H86 and the amide proton of L61 in NS1 effector domain (PDB ID:2KKZ). As a
result, L61(z-score =6.66,$\delta_H$ = 12.66 ppm) amide chemical shift is strongly downfield shifted.
**3.4 Bias, Structure, and Dynamics**
Potential bias in the BMRB and PDB data likely undercounts the occurrence of aromatic hydrogen bonds. Absent assigned NOEs,
the likelihood that an NMR structure will reflect a hydrogen bond to an pi cloud of an aromatic ring is low, because the additive



force fields used to refine most NMR structures, such as X-PLOR/CNS, do not capture the favorable interaction energy. To explore
the Van der Waals interactions in an H-bonding geometry, we used MoSART(Hoch and Stern, 2003) to simulate ALA approaching
PHE with the amide N-H of the former exactly aligned with the ring normal of the latter. The AMBER99 force field(Wang et al.,
2000) was used to compute the energy.

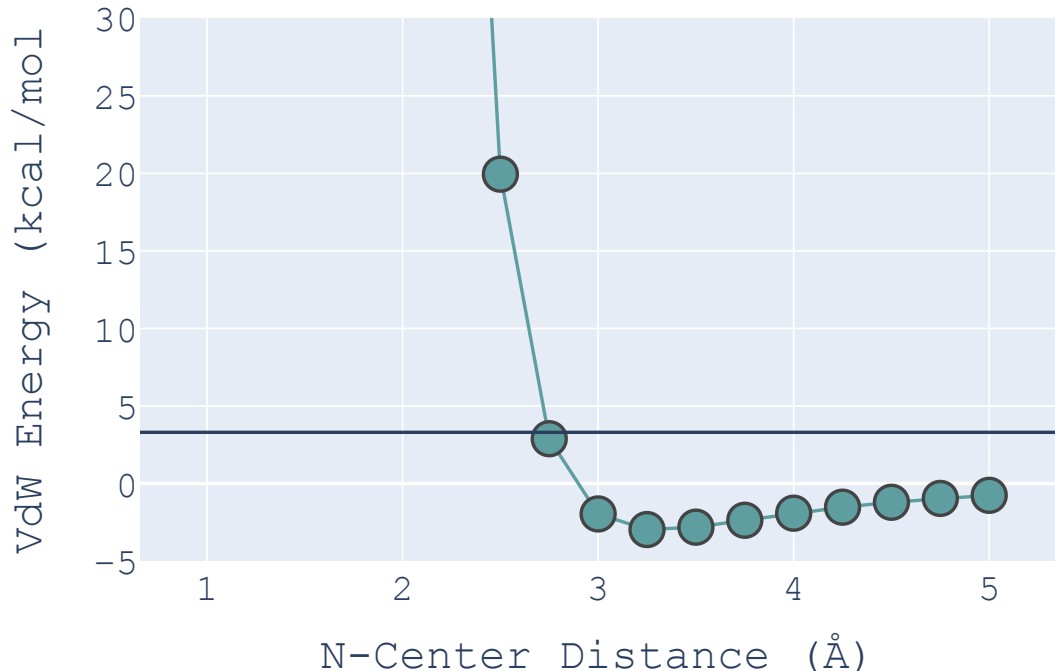


**Figure 9:** *Van der Waals interaction energies for ALA approaching PHE with its amide N-H aligned with the ring normal. On the*
*x-axis is the distance from the ALA nitrogen to the PHE ring center. VdW interaction energies for each distance were calculated*
*by subtracting the VdW energies of ALA and PHE in isolation from the energies calculated at that distance from one another. All*
*calculations were performed in MoSART using AMBER99 force fields.*


The results, shown in Figure 9, agree with those presented by Levitt and Perutz(Levitt and Perutz, 1988): there is a local minimum
in the van der Waals (VdW) energy with the amide nitrogen 3.3 Å  from the ring center.  The calculations also show that the non-
bonded VdW interactions do not preclude adoption of a hydrogen-bonded aromatic ring, however the well depth is so small that
the VdW attraction alone is likely insufficient to yield a favorable H-bond geometry without additional restraints.
Lack of assignments are not evidence of the absence of an NOE. Missing assignments (for example, 6280 out of 8111 outlying
amide proton shifts($|Z|>2$) do not have assigned NOEs to an aromatic ring) also would lead to an undercount. Possible bias in
BMRB notwithstanding, such as missing assignments not uniformly distributed, trends in shifts and NOE restraints for different





amino acid types that mirror one another provide a form of cross-validation and suggest that the shift outliers are not simply the
result of being buried in the protein and thus easier to assign. Bias in PDB NMR structures could reflect current practice in structure
refinement, which is dominated by restrained molecular mechanics simulations using empirical force fields augmented with
experimental restraint potentials. The forms of these restraint potentials can introduce bias (Hoch and Stern, 2005), and the additive
potentials that are used do not explicitly model p-π hydrogen bonds. Absent NOE or ring current restraints, NMR structures are
likely to under-represent aromatic hydrogen bonds.

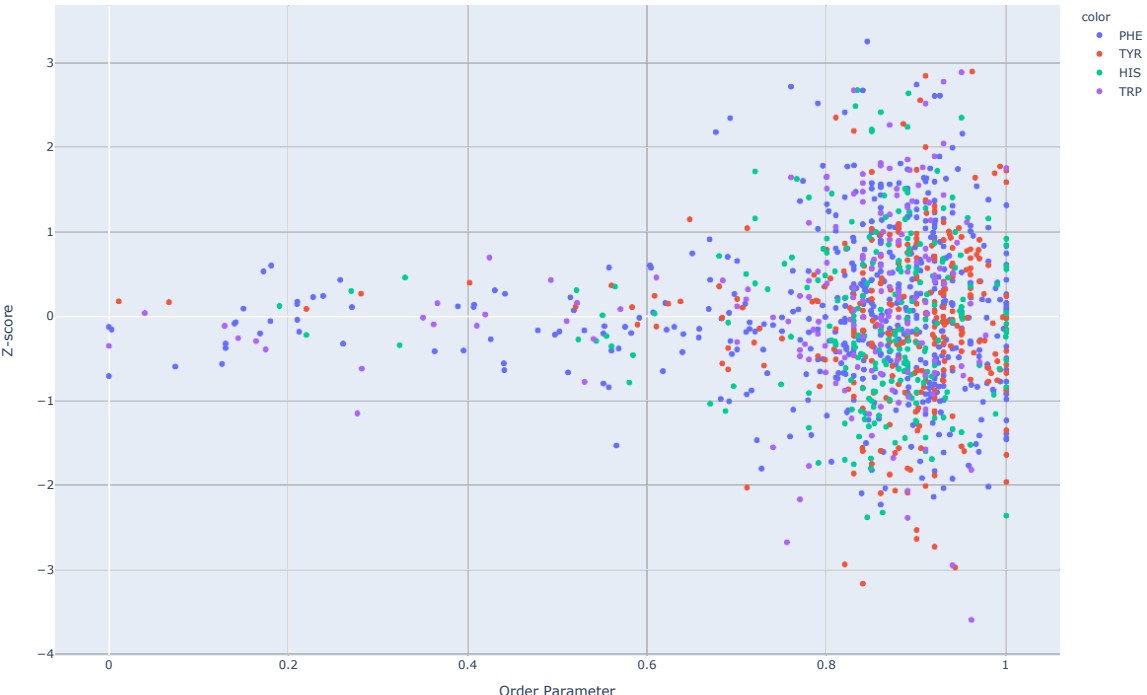


***Figure 10***. *Correlation of Z-scores with order parameters.*

In general, dynamics and disorder render chemical shifts toward their random-coil or median values (Dass et al., 2020; Nielsen
and Mulder, 2020). The correlation between secondary shift and order parameters is sufficiently strong that it has been used to
predict order parameters from chemical shifts (Figure 10). (Berjanskii and Wishart, 2005) Ring current effects in particular are
diminished by fluctuations about the $\chi_2$ torsion angle. (Hoch et al., 1982) Hydrogen bonds involving aromatic rings should diminish
these torsional fluctuations and should find correlates in side-chain relaxation properties for aromatic residues. Solution NMR
structures in general tend to be more flexible than crystal structures (Fowler et al., 2020), and inclusion of hydrogen bonding
interactions between amide groups and aromatic rings could reduce the flexibility and potentially improve the accuracy of NMR
structures.



Although chemical shifts have been used to refine protein NMR structures (Shen et al., 2009; Berjanskii et al., 2015; Cavalli et al.,
2007), for the most part these approaches leverage the influence of backbone torsion angles on chemical shifts, and do not consider
the influence of nearby sidechains. Despite evidence  that chemical shift refinement software is being used more frequently, the
pace of chemical shift-refined structure depositions remains low (Figure 11).

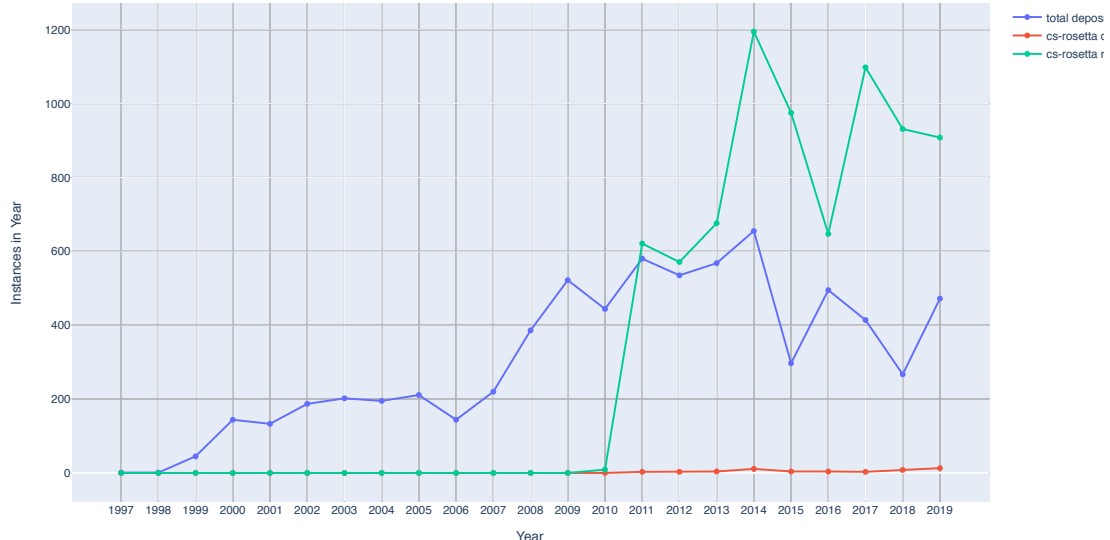


***Figure 11.*** *Trends in total BMRB structure depositions (blue)**,** runs executed using the BMRB CS-Rosetta server (green), and*
*depositions citing CS-Rosetta (red).*

Filtering the data plotted in Figure 3 to include only structures that reference CS-Rosetta (Figure 12) does not alter the overall
distributions. A challenge confronting a deeper understanding of these effects is that the available metadata in BMRB does not
articulate workflows, for example whether CS-Rosetta is used to generate initial trial structures or as a final refinement step), nor
does it indicate when ring current shift restraints were utilized.



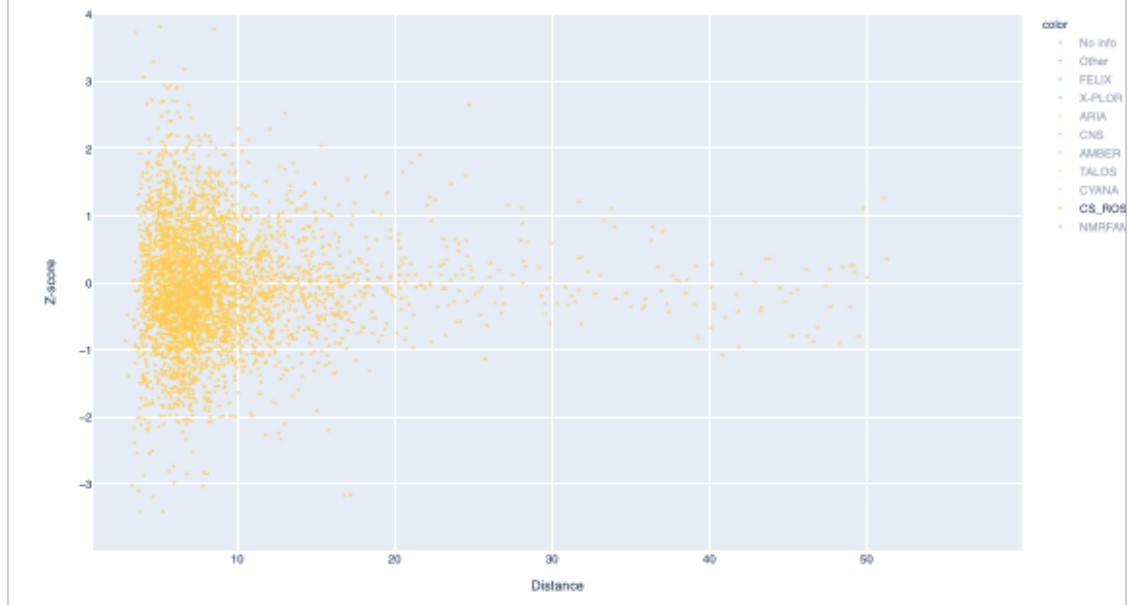


**Figure 12.** *The distribution of amide chemical shifts for depositions citing C S-Rosetta as function of distance from the center of the nearest ring (compare Figure 3).*

## 4 Concluding Remarks

Ring current shifts have a long history of providing structural insights from NMR studies of globular proteins (Perkins and Dwek, 1980), especially for methyl groups, whose secondary shifts tend to be dominated by ring current shifts. Early studies were largely anecdotal, focusing on individual proteins or small surveys. While relatively dynamic aromatic rings (for example Tyr and Phe rings that undergo ring flips on the fast exchange time scale) and disorder diminish the influence of ring current effects on secondary shifts (Hoch et al., 1982), the accumulation of data in BMRB for folded proteins has provided a wealth of amide chemical shifts exhibiting large secondary chemical shifts. Federation of BMRB chemical shift data with structural data from PDB confirms the strong correlation between proximity to an aromatic ring and extreme secondary shifts. Markedly different secondary shift trends for different aromatic residue types suggests promising avenues for improving proteins structure determination by NMR. Though chemical shift refinement has been repeatedly demonstrated (Perilla et al., 2017), it has not yet been widely adopted.

The extreme outlier amide chemical shifts and corroborating NOE effects examined here provide strong evidence of the widespread existence of amide-aromatic hydrogen bonds, but they are not fully conclusive. Nonetheless potential for under-representation in the BMRB data exists because of incomplete assignments. Relaxation studies on ring dynamics, contrasting rings where evidence suggests the presence of hydrogen bonding with rings lacking such evidence, could provide additional corroboration. Molecular mechanics simulations and structure refinement using polarizable force fields could reveal additional aromatic hydrogen bonds and restricted ring dynamics in folded proteins. We have initiated investigations along some of these lines.

More broadly, this preliminary investigation highlights the potential for unlocking latent knowledge hidden in BMRB, PDB, and other biological databases. The challenges posed include curation and validation of the data repositories and federation of data



between repositories. Robust and efficient solutions to these challenges are needed in order to realize the full promise of emerging
methods in Machine Learning. (Hoch, 2019)

## 5 Acknowledgements

This work was supported by a grant from the Miriam and David Donoho Foundation, and by grants from the US National Institutes
of Health (R01GM109046; P41GM111135) and from the University of Connecticut Office of the Vice President for Research
(CARIC). We thank Milo Westler and Charles Schwieters for helpful discussions.



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
