# Peer review of "Anomalous Amide Proton Chemical Shifts as Signatures of Hydrogen Bonding to Aromatic Sidechains"

_Magnetic Resonance, 2021_

## Referee Comment (RC1)

**Anomalous Amide Proton Chemical Shifts as Signatures of Hydrogen Bonding to Aromatic Sidechains**

Kumaran Baskaran, Colin W. Wilburn, Jonathan R. Wedell, Leonardus M. I. Koharudin, Eldon L. Ulrich, Adam D. Schuyler, Hamid R. Eghbalnia, Angela M. Gronenborn, and Jeffrey C. Hoch

General Comments

The authors have searched the Biological Magnetic Resonance Bank (BMRB) for unusual amide proton chemical shifts and amide-aromatic nuclear Overhauser effects of proteins. For each amide proton, they checked at the same time the Protein Data Bank (PDB) for the distances to the nearest aromatic protons of sidechain residues, i.e. tyrosine (TYR), phenylalanine (PHE), tryptophane (TPR) and histidine (HIS). They prepared a huge dataset consisting of 363686 amide protons from 467070 entries. The data show that the proximity of an amide proton to an aromatic ring can lead to a low-field or an upfield or to no signal discplacement with respect to the normal amid chemical shift of the residue of interest. When the azimuthal angle $\theta$ between the NH distance vector and the axis perpendicular to the aromatic plane is smaller than the magic angle, high-field shifts and at larger angles low-field shifts result.

Firstly, the distance effects are discussed. Surprisingly, Figure 3 indicates that aromatic induced shifts decay to 1/e only at distances around 30 Å. The shortest distances to the aromatic rings observed are surprisingly small, i.e. around 2.2 Å for TYR. For TYR, PHE and TPR the upfield shifts are more pronounced than the low-field shifts. By contrast, for amide NH interacting with HIS there is a low-field shifts preponderence. The azimuthal angles consistent with the chemical shifts are discussed in detail, as well as NOE distance constraints of amide NH groups to arometic rings. The authors interpret their data in terms of hydrogen bonds of the amide NH groups to the aromatic $\pi$ electrons, dominating in the case of TYR, PHE and TRP, whereas they discuss amide NH hydrogen bonds to the non-protonated nitrogen atoms of neutral histidine residues.

Finally, two examples are discussed, and force-field calculation are used to discuss the energetics of the interactions.

In summary, the article contains interesting insights into the interactions of amino NH groups with the arometic residues of side chains. I have the following comments, which I hope help the authors to improve the manuscript.

Specific Comments

(1) I had problems with the approach of the authors. As the amide NH chemical shifts probably refer to aqueous solution and the distances to crystal structures taken from the PDB which also contains NMR structures, I would have expected two Figures 3, one with the distances derived from NMR structures and the other from crystals. Or did I overlook something? Weak interactions might change between the liquid and the solid state. A discussion of the origin of the distances will be helpful.

(2) My main scientific concern is the use of the term "hydrogen bonding". The IUPAC definition (doi:10.1351/PAC-REC-10-01-02) is "*The hydrogen bond is an attractive interaction between a hydrogen atom from a molecule or a molecular fragment X–H in which X is more electronegative than H, and an atom or a group of atoms in the same or a different molecule, in which there is evidence of bond formation.*" According to the force-field calculations of an alanine-phenylalanine pair presented in Figure 9, a NH…$\pi$ interaction is found exhibiting an energy minimum at a N-ring distance of 3.3 Å. For the benzene-$NH_3$ complex in the gas phase a distance of 3.59 Å (Rodham et al. Nature **1993**, *362*, 735) and 3.54 Å (Mons et al. PCCP, **2002**, *4*, 571–576 was found which is in good agreement. That leaves for the ring-H distance values between 2.3 and 2.6 Å. But then only a small part of the data in Figure 3 can refer to NH…$\pi$ H-hydrogen bonds. That means that - with some exceptions - other forces may be responsible for most of the findings of the authors. However, the approch of the authors provides a diagnostic tool of for NH…$\pi$ formation: observation of a

small NH-ring distance is not a sufficient criterium for such a bond, but a positive z-value is needed in order to exclude for example, a normal H-bond to a histidine nitrogen. It would be interesting, if the authors could "disentangle" the plots in Figure 3, and associate different interaction types to the different parts. Perhaps they should use a term like "extended NH…$\pi$ H-bond".

(3) I have problems to believe in a normal interation according to Figure 1 in the distance range above 10 Å extending to 30 Å and more between the interacting rings and NH groups. There is matter in between, perhaps water, the PDB will show that, I doubt that the magnetic interaction will go unaffected through that matter. Figure 1 refers in my opinion to the vacuum. Thus, the magnetic susceptibility needs to be taken into account, and that quantity is heterogeneous.

(4) The Z-core of amino groups is defined in eq 1. That quantity is plotted in the ordinate of Figure 3. In order to access rapidly the ordinate values the authors should discuss eq 1, give a feeling for the values of $\delta_{res}$, $<\delta_{res}>$ and $\sigma_{res}$ of amino acids. Perhaps a Table will be helpful.

(5) In the examples of Figures 7 and 8 the distance and azimuthal angles should be mentioned in the graph or the caption. In addition, not only the Z values, but also the values of $\delta_{res}$ (labeled differently in the caption as $\delta_H$), $<\delta_{res}>$ and $\sigma_{res}$ should be included.

(6) I had difficulties with the coordinate systems of Figures 3 and 4. The dimensions in the x-axis are lacking. The white help lines in the background are broad that it is almost impossible to determine the distance of shift of a given data point by increasing the size of the graph.

---

## Author Response (AR1)

**Anomalous Amide Proton Chemical Shifts as Signatures of Hydrogen Bonding to Aromatic Sidechains**

Kumaran Baskaran, Colin W. Wilburn, Jonathan R. Wedell, Leonardus M. I. Koharudin, Eldon L. Ulrich, Adam D. Schuyler, Hamid R. Eghbalnia, Angela M. Gronenborn, and Jeffrey C. Hoch

Reviewer 1

General Comments
The authors have searched the Biological Magnetic Resonance Bank (BMRB) for unusual amide proton chemical shifts and amide-aromatic nuclear Overhauser effects of proteins. For each amide proton, they checked at the same time the Protein Data Bank (PDB) for the distances to the nearest aromatic protons of sidechain residues, i.e. tyrosine (TYR), phenylalanine (PHE), tryptophane (TPR) and histidine (HIS). They prepared a huge dataset consisting of 363686 amide protons from 467070 entries. The data show that the proximity of an amide proton to an aromatic ring can lead to a low-field or an upfield or to no signal discplacement with respect to the normal amid chemical shift of the residue of interest. When the azimuthal angle $\theta$ between the NH distance vector and the axis perpendicular to the aromatic plane is smaller than the magic angle, high-field shifts and at larger angles low-field shifts result.
Firstly, the distance effects are discussed. Surprisingly, Figure 3 indicates that aromatic induced shifts decay to $1/e$ only at distances around 30 Å.

The shifts plotted are not solely due to ring current shifts. The distance dependence likely also reflects other shift effects that correlate with distance from the protein surface, not related to ring current effects. This may help explain the surprising long-range dependence.

The shortest distances to the aromatic rings observed are surprisingly small, i.e. around 2.2 Å for TYR. For TYR, PHE and TPR the upfield shifts are more pronounced than the low-field shifts. By contrast, for amide NH interacting with HIS there is a low-field shifts preponderence. The azimuthal angles consistent with the chemical shifts are discussed in detail, as well as NOE distance constraints of amide NH groups to arometic rings. The authors interpret their data in terms of hydrogen bonds of the amide NH groups to the aromatic $\pi$ electrons, dominating in the case of TYR, PHE and TRP, whereas they discuss amide NH hydrogen bonds to the non-protonated nitrogen atoms of neutral histidine residues.
Finally, two examples are discussed, and force-field calculation are used to discuss the energetics of the interactions.
In summary, the article contains interesting insights into the interactions of amino NH groups with the arometic residues of side chains. I have the following comments, which I hope help the authors to improve the manuscript.

Specific Comments
(1) I had problems with the approach of the authors. As the amide NH chemical shifts probably refer to aqueous solution and the distances to crystal structures taken from the PDB which also contains NMR structures, I would have expected two Figures 3, one with the distances derived from NMR structures and the other from crystals. Or did I overlook something? Weak interactions might change between the liquid and the solid state. A discussion of the origin of the distances will be helpful.

The data in figure 3 is derived from NMR structures, not crystal structures. Crystal structures (with only a handful of exceptions) don't have chemical shifts and thus lack Z-scores.

It has long been noted that NMR structures differ qualitatively from X-ray crystal structures by being less compact (Clore and Gronenborn; Williamson). The reasons for this are multifaceted. Crystal structures these days are typically determined at low temperatures (and proteins have a well-established thermal expansion;

Frauenfelder). Most importantly, packing forces that are not present in solution can play various structure-determining roles. In contrast, NMR structures are not uniquely determined by the experimental data, especially early ones and those at low resolution, and the empirical potential energy functions used in NMR structure determination can play a more prominent role than for crystal structures.

(2) My main scientific concern is the use of the term "hydrogen bonding". The IUPAC definition (doi:10.1351/PAC-REC-10-01-02) is "*The hydrogen bond is an attractive interaction between a hydrogen atom from a molecule or a molecular fragment X–H in which X is more electronegative than H, and an atom or a group of atoms in the same or a different molecule, in which there is evidence of bond formation*." According to the force-field calculations of an alanine-phenylalanine pair presented in Figure 9, a NH...π interaction is found exhibiting an energy minimum at a N-ring distance of 3.3 Å. For the benzene-$NH_3$ complex in the gas phase a distance of 3.59 Å (Rodham et al. Nature **1993**, *362*, 735) and 3.54 Å (Mons et al. PCCP, **2002**, *4*, 571–576 was found which is in good agreement. That leaves for the ring-H distance values between 2.3 and 2.6 Å. But then only a small part of the data in Figure 3 can refer to NH...π H-hydrogen bonds. That means that - with some exceptions - other forces may be responsible for most of the findings of the authors. However, the approch of the authors provides a diagnostic tool of for NH...π formation: observation of a small NH-ring distance is not a sufficient criterium for such a bond, but a positive z-value is needed in order to exclude for example, a normal H-bond to a histidine nitrogen. It would be interesting, if the authors could "disentangle" the plots in Figure 3, and associate different interaction types to the different parts. Perhaps they should use a term like "extended NH...π H-bond".

The shift measurements don't provide insight into the nature of the energetic interaction, they merely help confirm proximity and orientation. In principle the distributions could be used to compute a potential of mean force, which could give insight into the energetics, however the discordance between predicted and measured chemical shifts for many atoms suggest that details of aromatic side-chain packing remain imprecise.

(3) I have problems to believe in a normal interation according to Figure 1 in the distance range above 10 Å extending to 30 Å and more between the interacting rings and NH groups. There is matter in between, perhaps water, the PDB will show that, I doubt that the magnetic interaction will go unaffected through that matter. Figure 1 refers in my opinion to the vacuum. Thus, the magnetic susceptibility needs to be taken into account, and that quantity is heterogeneous.

The parameterization of the aromatic ring current shift factors for different aromatic amino acid side chains implicitly accounts for the magnetic susceptibility of the protein interior. Undoubtedly this susceptibility is inhomogeneous and different locally, however, implicitly assuming homogenity has enabled good agreement between measured 1H ring current shifts and ring current shifts predicted from high-resolution protein structures. (Hoch, 1983)

(4) The Z-core of amino groups is defined in eq 1. That quantity is plotted in the ordinate of Figure 3. In order to access rapidly the ordinate values the authors should discuss eq 1, give a feeling for the values of $\delta_{res}$, $<\delta_{res}>$ and $\sigma_{res}$ of amino acids. Perhaps a Table will be helpful.

These statistics are available on the BMRB web site (https://bmrb.io/ref_info/stats.php?restype=aa&set=full). We will note this in the revised manuscript.

(5) In the examples of Figures 7 and 8 the distance and azimuthal angles should be mentioned in the graph or the caption. In addition, not only the Z values, but also the values of $\delta_{res}$ (labeled differently in the caption as $\delta_H$), $<\delta_{res}>$ and $\sigma_{res}$ should be included.

We will revise multiple figures to address these and other suggestions.

(6) I had difficulties with the coordinate systems of Figures 3 and 4. The dimensions in the x-axis are lacking. The white help lines in the background are broad that it is almost impossible to determine the distance of shift of a given data point by increasing the size of the graph.

Reviewer 2

This is an interesting work with an important message about the abundance of NH...pi hydrogen bonds. I like the "big data" approach used by the authors, but I have a couple of comments that are perhaps worth addressing in the manuscript.

There is no doubt that XH...pi hydrogen bonds exist in abundance in all kinds of molecular crystals. Nevertheless, it will probably remain a debate topic whether shielding of a proton or even appearance of J-coupling is necessary a result of an overall attractive interaction. Alternatively, steric contraction (packing effects) could create a short but at the end energetically unfavorable XH...pi contact (or energetically "neutral" contact), still leading to H shielding and through-space J-couplings. Perhaps it is worth pointing out in the manuscript that in the presented flow of logic NH...pi interactions do not have to be attractive. The conclusions about the abundance of particular NH/pi spatial orientations will be valid regardless.

We agree, there is nothing in the NMR data that proves the interaction is attractive, however Occam's Razor would imply an attractive interaction as the simplest explanation for the non-random distributions observed. In light of some of our findings, we hope that computational chemists will re-visit the energetics of aromatic interactions with NH groups.

Which brings me to the second point of criticism, which is the choice of 8 A as a cut-off for N...H distances. If one wants to focus on H-bonds in particular and not on magnetic anisotropy effects in general, a more realistic cut-off of ca. 4 A would be desirable. I would guess that the presented data stretching all the way to 8 A contains only a small fraction of truly attractive interactions.

We chose the longer cutoff to illustrate the non-random nature of the distance/orientation distribution. As noted in response to reviewer1, the long tail of the distribution reflects additional effects, beyond those caused by ring currents.

It is worth noting that hydrogen bonds between NH groups and aromatic systems do not have to directed towards the hydrogen bond center. Instead, as it occurs in H-bonds to double and triple bonds, the NH group might be pointing towards one of the CC bonds. In the center of an aromatic ring with large electron-withdrawing effects there might be even a pi-hole (insufficient electron density), "repulsing" NH proton donor. I would guess that H-bonds to CC bonds will be characterized by larger values of azimuthal angles and also maybe be smaller effects on chemical shifts (because deshielding due to H-bonding will be better compensated by the ring current effects, which are diminished at the border between shielding and deshielding regions of the aromatics). Would it be possible to comment on the possibility to distinguish between H-bonds to the center and to the side of the aromatic ring?

This is an interesting and important point. Indeed, in the plots in Figure 4 one can notice a peak near 25° for Phe and Tyr, which is the azimuthal expected for amides directly above one of the ring carbons and 3.4 Å from the ring plane. We thank referee Peter Tolstoy for this insight.

Minor point:

Fig. 3 - the units of distances should be given in abscissa. Also it should be written explicitly in the caption which distance is plotted (between which atoms?).

Agreed.